# Importance of Agility Performance in Professional Futsal Players; Reliability and Applicability of Newly Developed Testing Protocols

**DOI:** 10.3390/ijerph16183246

**Published:** 2019-09-04

**Authors:** Damir Sekulic, Nikola Foretic, Barbara Gilic, Michael R. Esco, Raouf Hammami, Ognjen Uljevic, Sime Versic, Miodrag Spasic

**Affiliations:** 1University of Split, Faculty of Kinesiology, 21000 Split, Croatia (D.S.) (N.F.) (B.G.) (O.U.) (S.V.); 2Department of Kinesiology, University of Alabama, Tuscaloosa, AL 35487, USA; 3Research Unit, Education, Motricity, Sport and Health, UR15JS01, High Institute of Sport and Physical Education of Sfax, University of Sfax, Sfax 3000, Tunisia

**Keywords:** conditioning capacities, validity, team sports, dribbling, sport-specific test

## Abstract

The purpose of this study was to determine the inter- and intra-testing reliability of newly developed tests of the change of direction speed (CODS) and reactive agility (RAG) in competitive futsal players. Additionally, the developed tests were evaluated for their validity with regard to the differentiation of two performance-levels. Thirty-two professional male futsal players (age = 26.22 ± 5.22 years; body height = 182.13 ± 5.99 cm, body mass = 77.43 ± 8.00 kg) participated in the study. The sample was divided into two groups based on their level of futsal performance: A top-level-group (*n* = 12) and a team-level-group (*n* = 20). The variables included body height, mass, body mass index, a sprint over a 10-m distance (S10M), and eight newly developed futsal specific CODS and RAG tests. The CODS and RAG tests were performed by dribbling the balls (CODS_D and RAG_D) and without dribbling (CODS_T and RAG_T), and the performances on the dominant and non-dominant sides were observed separately. All CODS, and RAG tests performed on dominant side and RAG_T tests performed on the non-dominant side had good inter-testing (CV = 5–8%; ICC = 0.76–0.89) and intra-testing (CV = 4–9%; ICC = 0.77–0.91) reliability. However, RAG_D performed on the non-dominant side was not reliable (ICC = 0.60, CV = 10%). The top-level-players outperformed the team-level-players in the CODS and RAG tests that involved dribbling (*t*-test: 4.28 and 2.40, *p* < 0.05; effect sizes (ES): 0.81 and 1.5, respectively), while the team-level players achieved better results in the CODS_T (*t*-test: 2.08, *p* < 0.05; ES: 0.60). The proposed CODS and RAG tests that involved dribbling over a 3.2-m distance, especially on the dominant side, appeared to be reliable, as well as valid for distinguishing the performance level in futsal players.

## 1. Introduction

In order to objectively evaluate task-specific fitness components, it is of upmost importance to use tests which appropriately mimic the real-world situation in which the tested fitness component is challenged [1]. In competitive sport settings, this approach is evidenced throughout the development and utilization of sport-specific testing protocols [2,3]. In general, such tests are generally accepted to better assess the physical capacities related to successful performance in a given sport compared to general fitness tests [4,5,6]. Therefore, sport-specific tests that simulate basic movement patterns in real-play situations should be utilized among the majority of competitive athletes.

Agility has been defined as “a rapid whole-body movement with change of speed or direction in response to a stimulus” [7]. This quality is recognized as an important component of efficacy in various tactical duties, and particularly, as an important facet of effectiveness in team sports [8,9,10,11]. Indeed, for some team sports, such as basketball, soccer or handball, agility is recognized as one of the most important determinants of overall success [12,13,14]. As a result, it is an extremely important variable in team-sports that should be regularly monitored and assessed [15,16,17].

Testing for agility is not as straight-forward as most practitioners and sport scientists believe. In fact, it is highly complex, as the differentiation between non-reactive agility and reactive agility (RAG) deserves special attention. Non-reactive agility involves an active change of direction speed (CODS). Consequently, the term “pre-planned agility” is regularly used for CODS, while RAG is often referred to as “non-planned agility” [13,18,19].

Futsal is a sport that involves intermittent periods of high-intensity physical effort over two, 20-min periods per game [20,21]. The sport requires sudden changes in movement patterns, fast sprints, and rapid decision-making to obtain or maintain ball possession, and the research shows that agility appears to be a vital component to successful play [22,23,24]. Furthermore, all three field playing positions (e.g., guards, wingers, pivots) must be able to effectively change their positions during the game, which actually accentuate the necessity for their agility performance. In other words, irrespective of their primary game specific tasks (i.e., defender—keeping the ball, making the smart passes; winger—sprinting and passing; pivot [topman]—attacking and scoring), all futsal players must possess exceptional agility with and without the ball [25,26].

However, studies specifically examining CODS and RAG protocols in futsal are limited. Soccer, on the other hand, is a very similar sport with a body of research existing on the subject. For instance, Bullock et al. (2012) demonstrated that a novel reactive motor skills test that measured sprint, passing and RAG in soccer players displayed good reliability [27]. Recently, Pojskic et al. (2018) showed high reliability and applicability of newly developed tests of soccer-specific CODS and RAG that included stop and go scenarios with ball kicking [12]. In one of the rare studies which examined the agility in futsal, Benvenuti et al. (2010) showed that a reactive visual stimuli agility test (RVS-T) was more suitable than CODS for measuring agility in both futsal and soccer field conditions [26].

Compared to soccer, futsal players perform more changes of direction during a game, while dribbling a smaller ball across a reduced court size [26]. For these reasons, both CODS and RAG may be especially important physical fitness parameters in futsal [25,26], and hence specific research within the sport is needed. However, no study so far has examined tests of agility-components that involved dribbling with the ball in this sport. This is particularly important as ball control while executing quick changes of direction in a reduced court size compared to other sports is a common characteristic of futsal [25,26]. The development of reliable and valid agility-based tests involving dribbling in futsal may assist professionals involved in this sport to choose appropriate testing protocols to evaluate players. The achievement on such tests can be used for selection of talented players, as well as the evaluation of training and conditioning processes.

Therefore, the aim of this study was to determine the inter- and intra-reliability of newly developed tests examining CODS and RAG in a sample comprised of professional futsal athletes. Additionally, the developed tests were evaluated for their construct validity with regard to the differentiation of two performance-levels of futsal players. It was hypothesized that the newly developed tests would be reliable testing procedures that would distinguish futsal players of two performance-levels, thus providing an insight into the measurement tools’ construct validity.

## 2. Materials and Methods

### 2.1. Study Design

In the initial phase of the investigation, the authors of the study collaborated with futsal experts to gather opinions regarding futsal-specific agility. The authors presented the basic idea, technical details, and theoretical boundaries of test execution (i.e., available equipment) and asked the experts to express their opinion about the most appropriate scenario which will be convenient for the purpose of testing of the futsal specific CODS and RAG for all field players (e.g., defenders, wingers, and pivots). The applied knowledge obtained allowed the development of relatively simple testing procedures that evaluated different agility components of the sport (please see Procedures section below for details).

This study comprised of a repeated measurement design in order to define the reliability of the newly developed tests. Additionally, throughout the cross-sectional design, this study compared basic anthropometric variables, sprinting-, CODS- and RAG-performances from two groups of futsal players of varying performance-levels in order to identify the construct validity of the applied tests. First, a randomly selected subgroup of 11 players performed the agility protocols on two separate testing occasions (e.g., test-retest) that were separated by 6 days to define inter-testing reliability. The intra-testing reliability was obtained from the results of all the participants involved in the study (n = 32). Finally, the groups based on performance (i.e., team-level and top-level; see further for more details about dividing into performance-levels) were compared on dependent variables (body mass, body height, sprinting-, CODS- and RAG-performance).

### 2.2. Participants

Thirty-two male professional futsal players (age = 26.2 ± 5.2 years; body height = 182.1 ± 6.0 cm, body mass = 77.4 ± 8.0 kg) from three futsal teams, competing at the highest national level in Croatia, voluntarily participated in the study. Although more players were originally tested (e.g., 35), in this study, the participants were selected based on the following criteria: Minimum 7 years of active involvement in futsal; older than 18 years of age; free from injury or illness; and have regularly performed standard training for at least three weeks prior. The goalkeepers were not included in this investigation. For the purpose of this study, the total sample was divided into two groups based on the performance level: Top-level players (12 players) and team-level players (20 players). The top-level players were those who met at least one of the three following criteria: (1) They were members of senior-level national futsal team over the last two years; (2) were members of the junior-level national futsal in the last competitive season (<18 years); (3) participated in the Union of European Football Association (UEFA) Futsal Champions League over the last two years, which is the highest competition level for futsal teams in Europe. The team-level players were those who were not grouped as the top-level players (please see previous criteria) and who were members of the teams participating at the highest national competitive level for the observed season.

During the timeframe of testing, the players participated in 6–7 training sessions and one official game each week. One specific session per week was exclusively dedicated to weight training, pre-habilitation, and exercises designed to improve aerobic-anaerobic endurance. The remaining sessions involved technical-tactical preparation (60% of training focus), combined with small sided drills and skill-based games (20%) and training games (20%).

The ethics board of the first author’s institution provided approval of the research experiment (Ethical Board Approval No: 2181-205-02-05-14-001). All the participants were informed of the purpose, benefits and risks of the investigation. The participants voluntarily took part in the testing after they provided written consent.

### 2.3. Procedures

The variables in this study included body mass (BM), body height (BH), 10-m sprint (S10M), as well as the newly developed futsal-specific CODS- and RAG-tests. All the assessment methods followed standardized procedures and were performed with calibrated equipment by experienced evaluators.

For the S10M test, the participants were placed 1m behind the trigger line with their body leaned forward. The first timing gate (Powertimers 300, Newtest Oy, Oulu Finland, Core serial number: 08310013) remained on the trigger line (0 m), and the second timing gate was placed at the finish line (10 m). Each timing gate had reflecting sensors at 1 m height. The participants were told not to include backward movements at the start and to sprint at maximal speed the whole distance with avoiding a dive finish. Three sprint trials with a rest period of 2 min between each were performed. The inter-trial reliability based on the results of three attempts was high (Intraclass correlation coefficient = 0.90), and the best achievement of each participant was used for the analyses.

The agility-components were tested by newly developed futsal-specific CODS and RAG tests. The performance during CODS and RAG followed two procedures: (1) The participants had to touch the ball at the precise moment a change-of-direction occurred (CODS_T and RAG_T, respectively); and (2) the participants dribbled a ball during the execution of each test (CODS_D and RAG_D, respectively). All tests had a Y shaped pattern with the distances specified in the Figure 1 (Figure 1a for tests which involved dribbling; Figure 1b for tests which involved ball touching). The timing for the RAG tests began when the participants crossed the initial infrared signal. At that moment, a hardware module lit one 30 cm high cones (A or B). As no prior indication was provided for the RAG tests, the participants had to quickly notice the specific lit and react accordingly. Thus, the RAG performance was non-planned. For the CODS tests, the participants had advanced knowledge on which cone will light up, and therefore were able to pre-plan the movement template.

For the RAG_D and CODS_D, the participants were instructed to dribble a ball (Figure 1a) to a marked circle on the ground in front of the designated cone. The participants left the ball within the circle and then changed their direction to run back to the starting line as quick as possible (Figure 1a). For the CODS_T and RAG_T, the participants had to run to the ball, which was placed in front of the cone, touch it with the sole of the foot and then run back through the infrared signal to stop the timer (Figure 1b). The RAG and CODS tests were performed over five trials with either the known scenario (for CODS_D and CODS_T), or unknown scenario/template (for RAG-D and RAG_T). In some previous investigations, the authors used live people and video as stimuli for the change of direction in RAG testing to assure more realistic testing procedure [28]. However, this study decided to use light-stimuli because it allowed standardized conditions for all players. Specifically, this was one of the first studies which included ball-dribbling in test execution, and therefore, this study tried to ensure a controlled testing environment.

For the purpose of this study, the performance on the dominant and non-dominant sides were analyzed. To determine the dominant and non-dominant side, the mean value for all B-cone performances (i.e., executions on the right side), and all A-cone performances (i.e., executions on the left side) for each participant and each executed test (e.g., CODS_D, CODS_T, RAG_D, and RAG_T) was calculated first. The performance side with the lower mean value was determined as the dominant side for each executed test (for each player separately). As a result, eight agility-performances were recorded as follows: RAG with dribbling performed on dominant- (RAG_DD), and non-dominant sides (RAG_DND); RAG with ball touching at the moment of change of direction performed on dominant- (RAG_TD), and non-dominant sides (RAG_TND); and the corresponding performances on each CODS test (CODS_DD, CODS_DND, CODS_TD, and CODS_TND, respectively).

The measuring of the CODS -and RAG-performances was performed by a hardware device based on an ATMEL micro-controller (model AT89C51RE2; ATMEL Corp, San Jose, CA, United States). A photoelectric infrared sensor (E18-D80NK) served as an external time triggering input, and light emitting diodes were used as outputs. The photoelectric infrared sensor (see Figure 1—IR) has a response time of less than 2 ms and a digital output signal. The sensor’s distance for detection ranged from 3 to 80 cm with the capability of detecting transparent objects. The sensor was connected with a microcontroller IO port (Figure 1—IR). The device was connected to a PC operated on Windows 7 operating system.

The testing was performed in an indoor facility on plastic turf where all participants regularly trained and competed. All players were tested at the same location at the approximately the same time of the day in order to avoid diurnal variations (between 16:00 and 19:00). The standardized warm up was performed before testing that included (in order): 5 min of light jogging; 5 min of combined lunges, jumps, one-leg hops, and changes of direction; and 5 min of dynamic stretching exercises. The testing was arranged in groups of 3–4 participants which allowed for the appropriate rest intervals between the tests and trials. A day before the data from CODS and RAG were collected, all players were familiarized with the procedures. The original testing field scenarios were applied, but players were instructed to perform each given test at a submaximal effort to identify the most appropriate movement pattern. Prior to the familiarization trials, each test was briefly and verbally presented to the participants by one of the authors of the study. All participants performed 2–3 familiarization trials of each test. The same day of familiarization, the participants were tested for sprinting capacity and anthropometrics variables. The CODS and RAG tests were performed at maximal effort for the data collection over the two days that followed familiarization. The CODS and RAG tests were performed in a random order, in which half of the sample performed CODS first and RAG second, and vice versa for the other half. All testing trials were performed after either protocol was initiated. The rest periods between each trial and each test was 2–3 min and 4–5 min, respectively. The trial for familiarization, sprint performance, and anthropometric data among the 11 players who performed the test-retest reliability procedures occurred 1 week prior to the other subjects. The CODS and RAG data collection for this cohort took place over the following two days in randomized order, according to the previously explained protocol. Then, this group was re-tested on CODS and RAG during the subsequent week, together with the other subjects.

### 2.4. Statistical Analysis

All the data were log-transformed to reduce the non-uniformity of error, and normality was tested using the Kolmogorov–Smirnov test procedure. The homoscedasticity for all variables was tested by Levene’s test. The descriptive statistics included the means and standard deviations presented as the true results for each variable (non-log-transformed).

The intraclass correlation coefficient (ICC) was used for analyzing the relative reliability, and the coefficient of variation (CV) was used for analyzing the absolute reliability [29]. The ICC was calculated from the variance estimates derived from a repeated measures analysis of variance, where the test and retest results were observed as repeated measures. The calculation of inter-testing ICC was done on the basis of the results of those participants who participated in two testing sessions (i.e., the 11 participants in the test-retest cohort).

Further, to define the intra-testing reliability (i.e., reliability across the trials in one testing session), the means and SDs for all trials and all study participants were used. A repeated measures analysis of variance over the testing trials with the corresponding Tukey post hoc test, were used to assess systematic errors (e.g., learning, fatigue) among the trials. Further, the intra-testing ICCs and CVs were calculated as previously suggested [30,31]. The ICC ≥ 0.75 and CV ≤ 10% were considered to indicate good (appropriate) reliability of studied CODS and RAG tests [32,33,34].

The Pearson’s product moment correlation coefficients were calculated to define the associations between RAG and CODS performances.

The construct validity of the tests was assessed by comparing performance groups in each test. The differences between performance-groups were calculated using 2-sided *t*-test for independent samples. Additionally, the effect sizes (ES; Cohen’s d) for the differences in sprinting- and agility-performances between the performance-group were calculated, and interpreted using the following qualitative descriptors: <0.2 as trivial, 0.21–0.49 as small, 0.50–0.79 as medium, >0.79 as strong [35].

## 3. Results

The demographic and anthropometric status for the studied players is presented in Table 1. The performance groups did not significantly differ in age and the anthropometric indices.

The inter-testing reliability of the newly developed tests of agility-performances are presented in the Table 2. In brief, the reliability of all CODS-based tests was good, with lowest ICC parameters evidenced for CODS_DND (ICC = 0.79), and highest for CODS_TD (ICC = 0.89). The reliability of RAG procedures was somewhat lower, and ranged from low ICC value for RAG_DND (ICC = 0.60), to good reliability in other RAG-based tests (ICC = 0.76–0.83).

The intra-testing reliability of the tests varied, with ICC values ranging from 0.69 for RAG_DND, up to 0.91 for CODS_DD and CODS_TD (Table 3). The absolute intra-testing reliability as indicated by CV showed the highest variation (e.g., lowest reliability) in test results for RAG_DD (9%), and the lowest variation (e.g., highest reliability) for two CODS test performed without dribbling (4%). Further, ANOVA showed significant differences among the testing trials of CODS_DND, while post-hoc differences for this test were statistically significant when the first trial was compared with the remaining two testing-trials, with no significant post-hoc differences between the 2nd and 3rd testing trial of this procedure.

The team-level group achieved significantly better results in CODS_TND (*t*-test: 2.08, *p* = 0.04, ES: 0.60) than top-level group. On the other hand, the top-level players outperformed the team-level players in CODS_DD (*t*-test: 2.40, *p* = 0.02, ES: 0.81), and RAG_DD (*t*-test: 4.28, *p* = 0.01, ES: 1.5) (Table 4; Figure 2).

While the correlations between the RAG and CODS tests performed without dribbling ranged from 0.09 (for correlation evidenced between RAG_TD and CODS_TD) to 0.51 (between CODS_TND and CODS_TD), the associations between the RAG and CODS performances which involved dribbling were considerably lower (Pearson r: 0.08-0.31). Collectively, futsal specific RAG and CODS tests shared <1%–25% of the common variance (Table 5).

## 4. Discussion

This study aimed to evaluate the reliability and construct validity of novel futsal-specific RAG and CODS tests. There were several noteworthy findings that should be highlighted. First, the reliability for most of the tests was considered good, with the exception of the RAG test that involved dribbling performed on the non-dominant side. Second, the complexity of the tests was a factor that altered the final achievement and the reliability of each agility test. Third, the CODS and RAG tests that involved dribbling were valid in distinguishing two performance-levels of futsal players. Therefore, our initial study hypothesis was partially approved.

### 4.1. Reliability of the Tests

The reliability for most of CODS and RAG tests studied in this investigation was appropriate. On the other hand, our tests had somewhat lower reliability than tests of similar capacities presented in previous futsal and soccer studies. For example, the RAG test that included dribbling had somewhat lower reliability than test of reactive visual stimuli agility in female futsal players (ICC = 0.76 vs. 0.80, respectively) [25]. Further, a complex agility protocol in soccer displayed better reliability (CV = 2.4%) than the current findings of reactive agility tests (CV = 6–10%) [26]. Last, Hachana et al. reported almost perfect reliability (ICC = 0.99) for a modified Illinois-CODS test conducted with 14 year old soccer players [36], while the current reliability results of CODS tests were evidently lower (ICC = 0.79–0.89, and 0.83–0.91 for inter-testing and intra-testing reliability, respectively).

In explaining such differences in the reliability between the current and previous findings, several issues should be noted. First, all the tests studied previously were characterized by non-stop movement patterns that involved changes of direction not performed from zero-velocity [25,26,36]. However, the agility-based tests presented herein involved more demanding movement templates of stop-and-go running. In addition, none of the tests in previous studies involved dribbling with the ball, and heterogeneous samples of participants were common. Since heterogeneity of the sample increases the variability of the results, consequently it increases the numerical values of the correlation coefficients (both between trials and/or measurements), and results in better reliability parameters [37].

The futsal specific CODS tests displayed better reliability than the RAG tests. Furthermore, the protocols which involved dribbling were of lower reliability than those without dribbling. Finally, the tests performed on the dominant side were more reliable than those performed on the non-dominant side. All of these differences in reliability are probably connected to the differences in text complexity (CODS vs. RAG, and with- vs. without-dribbling), and the participants’ familiarity with performance and execution (dominant vs. non-dominant side). First, CODS tests are known to be less complex than RAG tests since during the RAG testing, there is a clear moment of surprise in which the athlete must promptly react to visual stimuli by immediately performing a proper movement pattern [13,38]. Second, the tests which involve dribbling are more technically demanding than those performed without dribbling which causes greater variations among the trials and simply mathematically reduces the reliability [39]. Finally, the dominant-side in futsal is related to the side of the court the player is regularly positioned, and better motor-proficiency in specific directions, which altogether result in better stability of the performance and directly influences reliability (i.e., better reliability of tests performed on dominant side because of the better motor-proficiency) [38,40].

Therefore, the low reliability of the RAG_DND is most likely accentuated by the consequence of test-complexity (e.g., dribbling performed in RAG-circumstances), combined with the players’ non-familiarity with the movement templates performed on the non-dominant side. It is possible that additional familiarization trials, and/or an increased number of testing trials might have increased the reliability of RAG_DND, as it has been suggested previously for other fitness components [41]. However, it must be stated that futsal players frequently perform on one side of the court, and most of them use specialized movement templates that are almost exclusively executed on the dominant-side. While ecological validity and specificity are important considerations for any given test, the usefulness of the RAG testing in uncommon circumstances, such as during dribbling from the non-dominant side, is questionable.

### 4.2. Validity of the Tests

The CODS and RAG tests evaluated in this study shared less than 25% of the common variance. Logically, the correlation between corresponding tests was higher when testing procedures did not include dribbling. In general, the percentage of the common variance indicated that CODS and RAG capacities studied herein should be considered independent. Similar correlations between corresponding CODS and RAG tests were found among studies involving other team-sports [13,28,38], which supports the notion that planned and non-planned agility are influenced by different factors. For example, it has been suggested that sprinting speed and jumping capacities are important predictors of CODS. However, the perceptual-cognitive capacities and reactive-strength properties are suggested as being important factors influencing RAG-performances [42,43,44].

Only two testing procedures were considered valid as they relate to the applicability in distinguishing the participants by performance-level. Specifically, of the seven tests with good reliability, only CODS_DD, and RAG_DD showed superior scores in the top-level group. Meanwhile, other tests applied in this study, including the S10M, did not reveal better performance of advanced-level players. Therefore, it seems that the specific characteristics of futsal, with players constantly under the opponents’ pressure in a relatively small court, allow for the more complex and technically high demanding tests to differentiate the performance-levels [25]. The fact that team-players outperformed top-players in CODS_TND was not surprising since this performance was not highly related to skill-level (i.e., doesn’t include dribbling). Furthermore, the performance-groups did not differ in sprint capacity. It is possible that the team-level players are superior in other conditioning capacities known to be important determinants of CODS (i.e., jumping-, power-capacity) [34], which should be investigated in more detail in the future.

The current results contrast previous studies among other sports, as sport-specific CODS tests differentiated the performance better than RAG tests in groups of soccer and rugby-union players [12,45]. More precisely, in both previously mentioned studies, the larger ES differences between the performance-levels were evidenced for CODS, while in the current study the ES differences between the performance-groups were somewhat larger for RAG. It is possible that discrepancies in the applied tests, as well as the subject characteristics among the studies partially explain such small inconsistencies in the findings. For instance, considerably younger subjects were studied in the two previously mentioned studies that showed better construct validity of sport-specific CODS tests [12,45]. Therefore, it is possible that advanced age and sport-experience might contribute to the improvement in skill-level than the conditioning properties known to be important in CODS [14,34]. Logically, this would directly translate into better skill-based capacities, such as CODS and RAG with dribbling. Although further studies are needed in order to confirm the previous explanation, the lack of differences in sprinting capacities between the performance-groups indirectly support this discussion. Specifically, sprinting capacity is a clear conditioning capacity with a limited influence of the specific skill on test performance. Therefore, a lack of difference in S10M, together with the superiority of the top-level group in complex agility-based tasks indicates that the differences between the performance-levels in futsal are more related to the players’ skill-level than to conditioning status.

### 4.3. Limitations and Strengths

This study has several limitations. First, this study examined only male adult futsal players. Therefore, the generalizability of the results is limited to similar samples of participants. Furthermore, the testing procedures consisted of futsal specific movement templates, but no attention was directed towards position-specific tasks in the futsal game. Finally, apart from agility-components, the only other variables were body mass, height and sprinting capacity. Therefore, the information on other important fitness components are lacking. On the other hand, this is one of the first studies which developed and examined CODS and RAG tests with dribbling, which are known to be common real-game-demand in futsal. Moreover, the reliability of the tests was extensively evaluated while observing both intra- and inter-testing reliability of the developed measuring protocols. Therefore, this study’s results contribute to the knowledge in the field and initiate further investigations.

## 5. Conclusions

The results of the study showed that newly developed futsal-specific CODS and RAG tests were reliable. However, it is accentuated that proper reliability was achieved after specific familiarization of the players. In brief, a day before testing, all the players performed 2–3 familiarization trials at submaximal intensity in order to find the most appropriate movement template. Even after such familiarization, three maximal testing trials were necessary for the stabilization of the results.

The study highlighted the necessity of the specific evaluation of RAG and CODS capacities on the dominant and non-dominant side, especially with regard to the tests that involved ball dribbling. The coaches and practitioners working with futsal players should be informed of these results to make elaborated decisions concerning the most appropriate tests for their players. Specifically, the tests on the dominant side were generally reliable, and therefore useful for adult futsal players. However, the more complex tests that involved ball dribbling, or that were performed on the non-dominant side displayed lower reliability and hence require further evaluation.

The results suggested that differences between performance levels in futsal were related to the differences in skill-level. In fact, no differences between the groups existed in sprint capacity, and the team-level group even outperformed the top-level group in one of the studied CODS performance that was performed without dribbling. On the other hand, the top-level group was superior in CODS and RAG tests that involved dribbling, which demonstrated the importance of a sport-specific skill. Collectively, the findings point to the necessity of a combined evaluation of conditioning- and skill-capacities for a comprehensive approach towards optional athletic development.

## Figures and Tables

**Figure 1 ijerph-16-03246-f001:**
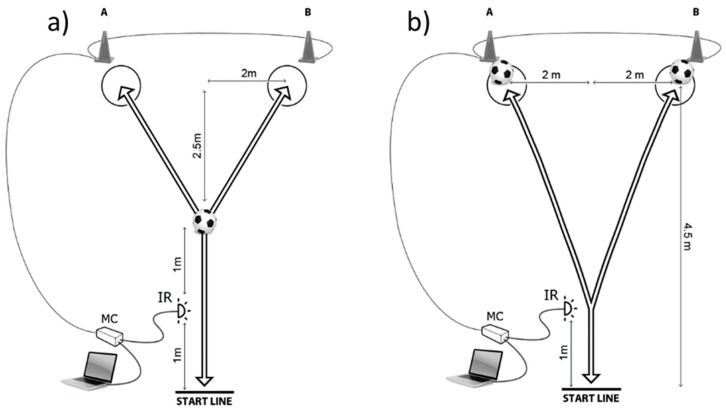
Testing of the futsal specific change of direction speed and reactive agility including (**a**) dribbling and (**b**) ball touching (MC—microcontroler, IR—infrared beam).

**Figure 2 ijerph-16-03246-f002:**
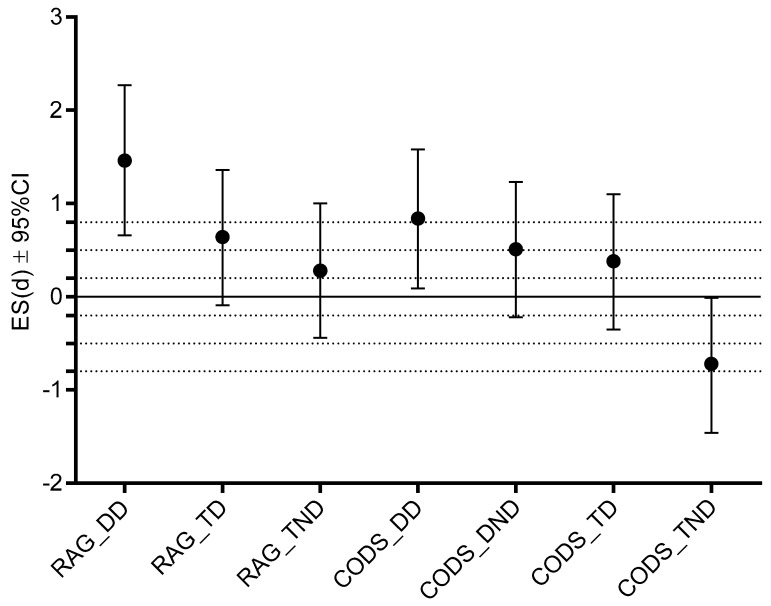
The effect size differences (ES; Cohen’s d) with 95% confidence intervals (95%CI) between the top-level and the team-level players (RAG_DD—reactive agility performed with dribbling on dominant side, RAG_TD—reactive agility performed with ball touching on the dominant side, RAG_TND—reactive agility performed with ball touching on the non-dominant side, CODS_DD—change-of-direction-speed performed with dribbling on dominant side, CODS_DND—change-of-direction-speed performed with dribbling on non-dominant side, CODS_TD—change-of-direction-speed performed with ball touching on dominant side, CODS_TND—change-of-direction-speed performed with ball touching on non-dominant side), <0.2 trivial ES, 0.21–0.49 small ES, 0.50–0.79 medium ES, >0.79 strong ES.

**Table 1 ijerph-16-03246-t001:** The demographic and anthropometric data of the futsal players included in two performance-groups.

Variables	Top-Level Players (*n* = 12)		Team-Players (*n* = 20)		*t*-Test	
	Mean	SD	Mean	SD	*t*-Value	*p*
Age (years)	25.8	5.6	26.9	5.1	−0.57	0.57
Body height (cm)	180.3	7.2	183.4	5.0	−1.46	0.16
Body mass (kg)	74.3	9.0	79.3	6.9	−1.76	0.09
BMI (kg/m2)	22.8	1.8	23.6	1.9	−1.14	0.26

**Table 2 ijerph-16-03246-t002:** Test–retest reliability results (n = 11).

Variables	Test 1		Test 2		Reliability	
	Mean	SD	Mean	SD	ICC	CV
RAG_DD (s)	2.54	0.14	2.50	0.15	0.76	0.06
RAG_DND (s)	2.80	0.11	2.40	0.15	0.60	0.10
RAG_TD (s)	2.41	0.14	2.38	0.11	0.83	0.08
RAG_TND (s)	2.51	0.19	2.45	0.15	0.77	0.07
CODS_DD (s)	2.35	0.18	2.35	0.22	0.81	0.05
CODS_DND (s)	2.56	0.19	2.59	0.18	0.79	0.05
CODS_TD (s)	2.06	0.12	2.05	0.13	0.89	0.07
CODS_TND (s)	2.33	0.10	2.32	0.13	0.81	0.06

LEGEND: RAG_DD—reactive agility performed with dribbling on dominant side, RAG_DND—reactive agility performed with dribbling on non-dominant side, RAG_TD—reactive agility performed with ball touching on dominant side, RAG_TND—reactive agility performed with ball touching on non-dominant side, CODS_DD—change-of-direction-speed performed with dribbling on dominant side, CODS_DND—change-of-direction-speed performed with dribbling on non-dominant side, CODS_TD—change-of-direction-speed performed with ball touching on dominant side, CODS_TND—change-of-direction-speed performed with ball touching on non-dominant side.

**Table 3 ijerph-16-03246-t003:** Intra-session reliability of the tests.

Variables	Trial 1		Trial 2		Trial 3		ICC	CV
	Mean	SD	Mean	SD	Mean	SD		
RAG_DD (s)	2.65	0.19	2.58	0.18	2.56	0.17	0.77	0.09
RAG_DND (s)	3.11	0.18	2.91	0.19	2.71	0.16	0.69	0.08
RAG_TD (s)	2.50	0.17	2.45	0.17	2.44	0.16	0.80	0.07
RAG_TND (s)	2.71	0.19	2.5	0.19	2.49	0.17	0.81	0.08
CODS_DD (s)	2.53	0.19	2.52	0.23	2.50	0.22	0.91	0.07
CODS_DND (s)	2.75	0.17	2.57	0.18	2.58	0.16	0.83	0.05
CODS_TD (s)	2.09	0.11	2.05	0.14	2.06	0.13	0.91	0.04
CODS_TND (s)	2.30	0.17	2.31	0.18	2.27	0.12	0.86	0.04

LEGEND: RAG_DD—reactive agility performed with dribbling on dominant side, RAG_DND—reactive agility performed with dribbling on non-dominant side, RAG_TD—reactive agility performed with ball touching on dominant side, RAG_TND—reactive agility performed with ball touching on non-dominant side, CODS_DD—change-of-direction-speed performed with dribbling on dominant side, CODS_DND—change-of-direction-speed performed with dribbling on non-dominant side, CODS_TD—change-of-direction-speed performed with ball touching on dominant side, CODS_TND—change-of-direction-speed performed with ball touching on non-dominant side.

**Table 4 ijerph-16-03246-t004:** Comparison between performance levels in the tested variables.

Variables	Top-Level Players(*n* = 12)		Team-Players (*n* = 20)		*t*-Test (df = 30)	
	Mean	SD	Mean	SD	*t*-Value	*p*
RAG_DD (s)	2.43	0.12	2.64	0.15	−4.28	0.00
RAG_TD (s)	2.38	0.17	2.48	0.14	−1.69	0.10
RAG_TND (s)	2.46	0.21	2.51	0.15	−0.70	0.49
CODS_DD (s)	2.39	0.19	2.57	0.22	−2.40	0.02
CODS_DND (s)	2.57	0.16	2.65	0.15	−1.32	0.20
CODS_TD (s)	2.03	0.11	2.08	0.14	−0.96	0.34
CODS_TND (s)	2.31	0.12	2.23	0.10	2.08	0.04
S10M (s)	1.63	0.07	1.69	0.11	−1.64	0.11

LEGEND: RAG_DD—reactive agility performed with dribbling on dominant side, RAG_TD—reactive agility performed with ball touching on dominant side, RAG_TND—reactive agility performed with ball touching on non-dominant side, CODS_DD—change-of-direction-speed performed with dribbling on dominant side, CODS_DND—change-of-direction-speed performed with dribbling on non-dominant side, CODS_TD—change-of-direction-speed performed with ball touching on dominant side, CODS_TND—change-of-direction-speed performed with ball touching on non-dominant side, S10M—sprint over 10 m distance.

**Table 5 ijerph-16-03246-t005:** Pearson product moment correlation coefficients with corresponding significance level (p) between the studied agility variables *.

	RAG_DD	RAG_TD	RAG_TND	CODS_DD	CODS_DND	CODS_TD	CODS_TND
RAG_DD	-						
RAG_TD	0.09 (0.64)	-					
RAG_TND	0.35 (0.06)	0.01 (0.99)	-				
CODS_DD	0.59 (0.01)	0.52 (0.01)	0.12 (0.53)	-			
CODS_DND	0.45 (0.01)	−0.04 (0.84)	0.09 (0.61)	0.23 (0.20)	-		
CODS_TD	0.42 (0.02)	−0.09 (0.62)	0.20 (0.28)	0.24 (0.18)	0.72 (0.01)	-	
CODS_TND	0.02 (0.93)	−0.08 (0.63)	0.36 (0.04)	0.03 (0.87)	0.42 (0.02)	0.38 (0.04)	-
S10m	0.22 (0.22)	−0.09 (0.65)	0.34 (0.06)	0.06 (0.74)	−0.20 (0.27)	−0.02 (0.91)	−0.25 (0.17)

LEGEND: RAG_DD—reactive agility performed with dribbling on dominant side, RAG_TD—reactive agility performed with ball touching on dominant side, RAG_TND—reactive agility performed with ball touching on non-dominant side, CODS_DD—change-of-direction-speed performed with dribbling on dominant side, CODS_DND—change-of-direction-speed performed with dribbling on non-dominant side, CODS_TD—change-of-direction-speed performed with ball touching on dominant side, CODS_TND—change-of-direction-speed performed with ball touching on non-dominant side, S10M—sprint over 10 m distance.

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
