# Peer review of "Importance of Agility Performance in Professional Futsal Players; Reliability and Applicability of Newly Developed Testing Protocols"

_ijerph, 2019, doi:10.3390/ijerph16183246_

Round 1

Reviewer 1 Report

The research theme is current and extremely important, both for training, for competition and for the validation of instruments for practical application.

The introduction is in line with the state of the art, the objective is well defined. The experimental design is well done. 

The methodology is well outlined and described in detail and can be replicated by other researchers. The statistical procedures are in agreement with the studied theme and with the objective of validation of the new protocols. 

The results are well presented. The discussion is based and compared with other sports because there is little information on futsal and therefore some different results from the literature. It is original and has practical implications. 

The authors were careful to point out the limitations of the study, which demonstrates the importance of this type of validation and procedures to perform. The conclusions are direct and according to the theme, where the authors refer to the need for further studies, but the originality of the theme and its application in futsal is of utmost importance.

My suggestion to the authors is that they can briefly buy the various positions in futsal, because although this sample is small, it is high level and the data for the various positions (pivot vs defense) can be an asset to the coaches. 

The bibliography is very current, apart from the classic bibliography according to the studied theme. 

Therefore, the study is an asset for research, especially for the quality of the methodology, originality in futsal, practical application in training and competition.

Author Response

REVIEWER 1

 The research theme is current and extremely important, both for training, for competition and for the validation of instruments for practical application.

The introduction is in line with the state of the art, the objective is well defined. The experimental design is well done.

The methodology is well outlined and described in detail and can be replicated by other researchers. The statistical procedures are in agreement with the studied theme and with the objective of validation of the new protocols.

The results are well presented. The discussion is based and compared with other sports because there is little information on futsal and therefore some different results from the literature. It is original and has practical implications.

The authors were careful to point out the limitations of the study, which demonstrates the importance of this type of validation and procedures to perform. The conclusions are direct and according to the theme, where the authors refer to the need for further studies, but the originality of the theme and its application in futsal is of utmost importance.

My suggestion to the authors is that they can briefly buy the various positions in futsal, because although this sample is small, it is high level and the data for the various positions (pivot vs defense) can be an asset to the coaches.

The bibliography is very current, apart from the classic bibliography according to the studied theme.

Therefore, the study is an asset for research, especially for the quality of the methodology, originality in futsal, practical application in training and competition.

RESPONSE: Thank you for recognizing the quality of our manuscript and work. Indeed, we tried hard to cover all necessary aspects of the research, especially with regard to methodology and sample selection. In the revised version of the manuscript we tried to additionaly improve the quality and readability of the papers following yours and suggestions of other two reviewers. Specifically, with regard to your suggestion about elaboration of „position specifics“ in futsal. We must agree that this issue was not properly covered in the original version of the manuscript, and therefore we are particularly grateful for your suggestion. Added text reads: “Futsal is a sport that involves intermittent periods of high-intensity physical effort over two, 20-minute periods per game [20-22]. Research shows that agility appears to be a vital component to successful play [23]. The sport requires sudden changes in movement patters, fast sprints, and rapid decision-making to obtain or maintain ball possession [24,25]. This is particularly emphasized by the fact that all three field playing positions (e.g. guards, wingers, pivots) must be able to effectively change their positions during the game, which actually accentuate the necessity for agility performance. In other words, irrespective of their primary game specific tasks (i.e. defender – keeping the ball, making the smart passes; winger – sprinting and passing; pivot [topman] – attacking and scoring) all futsal players must possess exceptional agility with and without the ball [26].” (Please see 4th paragraph of the Introduction).

Staying at your disposal for additional improvements.

Authors

Reviewer 2 Report

The authors have done an excellent job at preparing a clear and succinct manuscript exploring a relevant topic of CODS and reactive agility testing in futsal. I have raised some important points below that need to be address to improve the overall manuscript:

#1 Please check the grammar throughout the manuscript as there are instances of errors throughout in the writing. For example, on line 24, "CODS tests, RAG tests" could just be "CODS and RAG tests"; on line 30, "tests of CODS and RAG" should just be "CODS and RAG tests"; on line 48, "regarding" should be "in"; one line 58, "exceeded" should be "exceed"; on line 62, "patters" should be "patterns". More instances that need addressing are throughout.

#2 Abstract, line 28: Consider making a statement for tests without dribbling on this result here.

#3 Keywords: Try not to replicate words used in the title.

#4 Introduction, lines 43-44: Modern definitions identify "agility" as a change in direction/speed in response to a stimulus - the definition you provide is more so for change-of-direction speed. Please address and appropriately reference the definition you provide.

#5 Introduction, line 45: Agility is not a health-related fitness construct, it is purely performance-related, so I feel this term should be removed here.

#6 Introduction, lines 57-58: I am unsure what this sentence means (sentence beginning with "Specifically") and feel it should be deleted as it adds nothing to the section and is not appropriately referenced.

#7 Introduction, line 72: Please be specific throughout and not mention "agility" generally, but indicate whether you are referring to CODS or RAG.

#8 Introduction, line 83: Instead of identifying "differentiation of two performance-levels", why not explain or identify this as "construct validity" as most studies do in the literature? It is a form of validating the tests and you mention it on line 86, but I feel you could consistently mention this elsewhere also.

#9 Study design, line 89: But didn't you apply repeated measurements rather than solely a cross-section study to determine the tests' reliability? If so, you should adjust this sentence here.

#10 Study design, line 94: Delete "study's scientists to develop" and replace with "development of".

#11 Participants, line 102: Were the participants "selected" or did they volunteer? Please indicate clearly here.

#12 Participants, line 112: Were there any timeframes by which participants had to previously compete at these levels to fit this category (e.g. you could have somebody compete at this level 10 years ago, and be significantly lower now)? Also, what definition was used for "team-level" players?

#13 Procedures, line 130: 1 m behind the start line seems excessive - why was this distance chosen and can you cite with appropriate literature to support? Or do you mean 1 m behind the initial trigger light?

#14 Procedures, line 132: What is "intra-trial" reliability? Shouldn't this be "inter-trial" reliability given I assume you are looking at comparisons across repeated trials within the same session?

#15 Procedures, line 140: Why were light stimuli used in this study when they are not the most sport-specific stimuli to embed in such tests (e.g. live people or videos are more realistic and game-specific)? Please explain and justify further.

#16 Procedures, line 154: It is unclear what the 5 directions used were? Can you elaborate or make sure they are clearly explained in your methods?

#17 Procedures, lines 155-156: Has this approach been used to classify dominant and non-dominant previously? I am not convinced this is an ideal approach to use.

#18 Statistical analysis, line 200: Do you mean "inter-testing" here rather than "intra-testing"?

#19 Statistical analysis, line 208: What criteria was used to indicate a good CV? Please include.

#20 Statistical analysis, lines 212-214: Please delete your hypothesis here as it is not needed.

#21 Figure 2: The effect size for CODS TND should be a negative value (below zero) on this graph, given the team-level players performed better than top-level players. Please amend.

#22 Discussion, line 291: I believe you mean "exception" rather than "execution" here?

#23 Discussion, line 309: Please explain the impact of a heterogenous sample of participants on reliability more directly here.

#24 Discussion, line 340: The reason of "test complexity" has been repeated numerous times up to here and seems quite repetitive. Can you include alternative suggestions or perhaps group your statements on test complexity together more to avoid duplication?

#25 Conclusions: It might be worth highlighting how many trials are needed during familiarisation before consistent results are seen in specific tests either in this section or in your Discussion more directly.

Author Response

REVIEWER 2

The authors have done an excellent job at preparing a clear and succinct manuscript exploring a relevant topic of CODS and reactive agility testing in futsal. I have raised some important points below that need to be address to improve the overall manuscript:

RESPONSE: Thank you for recognizing the potential in our work. Also, we are particularly grateful to constructive comments and suggestions you have raised in your review. We tried to follow it specifically, and amended the manuscript accordingly. Please find bellow how we responded to your comments and where to find amended parts of the text.

#1 Please check the grammar throughout the manuscript as there are instances of errors throughout in the writing. For example, on line 24, "CODS tests, RAG tests" could just be "CODS and RAG tests"; on line 30, "tests of CODS and RAG" should just be "CODS and RAG tests"; on line 48, "regarding" should be "in"; one line 58, "exceeded" should be "exceed"; on line 62, "patters" should be "patterns". More instances that need addressing are throughout.

RESPONSE: Text is checked in details with regard to quality of the language and changes are done accordingly. Thank you.

#2 Abstract, line 28: Consider making a statement for tests without dribbling on this result here.

RESPONSE: Amended accordingly. Text now reads: “Top-level-players outperformed team-level-players in the CODS and RAG tests that involved dribbling (t-test: 4.28 and 2.40, p < 0.05) and the effect sizes of the differences were large (ES: 0.81-1.5), while team-level group achieved better results in the CODS_T (t-test: 2.08, p < 0.05; ES: 0.60).” (Please see highlighted text in Abstract section. Thank you)

#3 Keywords: Try not to replicate words used in the title.

RESPONSE: Amended accordingly. (e.g. conditioning capacities; validity; team sports, dribbling, sport-specific test)

#4 Introduction, lines 43-44: Modern definitions identify "agility" as a change in direction/speed in response to a stimulus - the definition you provide is more so for change-of-direction speed. Please address and appropriately reference the definition you provide.

RESPONSE: The definition provided by Sheppard and Young is used and text now reads: “Agility has been defined as “a rapid whole-body movement with change of speed or direction in response to a stimulus” [7].” (please see 1st sentence of the 2nd para in Introduction).

#5 Introduction, line 45: Agility is not a health-related fitness construct, it is purely performance-related, so I feel this term should be removed here.

RESPONSE: Removed as suggested.

#6 Introduction, lines 57-58: I am unsure what this sentence means (sentence beginning with "Specifically") and feel it should be deleted as it adds nothing to the section and is not appropriately referenced.

RESPONSE: Deleted as suggested

#7 Introduction, line 72: Please be specific throughout and not mention "agility" generally, but indicate whether you are referring to CODS or RAG.

REFERENCE: Text is amended and now reads: For these reasons, both CODS and RAG may be especially important physical fitness parameters in futsal [27] and hence specific research within the sport is needed.

#8 Introduction, line 83: Instead of identifying "differentiation of two performance-levels", why not explain or identify this as "construct validity" as most studies do in the literature? It is a form of validating the tests and you mention it on line 86, but I feel you could consistently mention this elsewhere also.

RESPONSE: Thank you. We defined “construct validity” and used this term later in manuscript. text reads: “Additionally, the developed tests were evaluated for their construct validity with regard to differentiation of two performance-levels of futsal players. (Please see last paragraph of the Introduction).” Please see last paragraph of the Introduction

#9 Study design, line 89: But didn't you apply repeated measurements rather than solely a cross-section study to determine the tests' reliability? If so, you should adjust this sentence here.

RESPONSE: Corrected, thank you. Text reads: “This study comprised repeated measurement design in order to define reliability of the newly developed tests. Additionally, throughout cross-sectional design we compared basic anthropometric variables, sprinting-, CODS- and RAG-performances from two groups of futsal players of varying performance-levels in order to identify construct validity of the applied tests. (Please see 2nd paragraph of the Materials and methods).

#10 Study design, line 94: Delete "study's scientists to develop" and replace with "development of".

RESPONSE: Amended accordingly. Thank you!

#11 Participants, line 102: Were the participants "selected" or did they volunteer? Please indicate clearly here.

RESPONSE: Indeed, they were volunteers. It is corrected, thank you.

#12 Participants, line 112: Were there any timeframes by which participants had to previously compete at these levels to fit this category (e.g. you could have somebody compete at this level 10 years ago, and be significantly lower now)? Also, what definition was used for "team-level" players?

RESPONSE: Thank you for noticing it. Indeed, we did not pay attention on “time-frame”, but mostly because this was not the problem since all top-level players were “fresh” in their performance-achievement. However, it is now specifically explained. Text reads: “Top-level players were those who met at least one of the three following criteria: 1) were members of senior-level national futsal team over the last two years; 2) were members of the junior-level national futsal in the last competitive season (<18 years); 3) participated in the Union of European Football Association (UEFA) Futsal Champions League over the last two years, which is the highest competition level for futsal teams in Europe. Team-level players were those who were not grouped as top-level players (please see previous criteria), who were members of teams participating at the highest national competitive level for the observed season.” (Please see 1st paragraph of the Participants subsection).

#13 Procedures, line 130: 1 m behind the start line seems excessive - why was this distance chosen and can you cite with appropriate literature to support? Or do you mean 1 m behind the initial trigger light?

RESPONSE: Thank you. We meant “behind the trigger line”. It is amended accordingly.

#14 Procedures, line 132: What is "intra-trial" reliability? Shouldn't this be "inter-trial" reliability given I assume you are looking at comparisons across repeated trials within the same session?

RESPONSE: Thank you. Amended accordingly.

#15 Procedures, line 140: Why were light stimuli used in this study when they are not the most sport-specific stimuli to embed in such tests (e.g. live people or videos are more realistic and game-specific)? Please explain and justify further.

RESPONSE: Yes, we are aware of the video-based testing procedures and its pluses, however, this study involved relatively complex skills (dribbling), and therefore we tried to achieve highly standardized conditions of testing for all participants (i.e. video based analysis depends of “reading of the specific body language” which is specific for each “video model”). It is now specified, and text reads: “Although is some previous investigation authors used “live people” and video as stimuli for change of direction in RAG testing (to assure more realistic testing procedure) [29], we decided to use light-stimuli because it allowed standardized conditions for all players (i.e. this was the first study which included ball-dribbling in test execution, and therefore we tried to assure controlled testing environment)” (Please see 3rd paragraph of the Procedures subsection) Thank you.

#16 Procedures, line 154: It is unclear what the 5 directions used were? Can you elaborate or make sure they are clearly explained in your methods?

RESPONSE: Indeed, the statement was not clear. It is amended, and hope it is clearer now. Text reads: “For the CODS tests participants had advanced knowledge on which cone will light up, and therefore were able to pre-plan the movement template. Because no prior indication was provided for the RAG tests, the participants had to quickly notice the specific lit and react accordingly. Thus, the RAG performance was non-planned. All tests were performed over five trials with either known scenario (for CODS_D and CODS_T), or unknown scenario/template (for RAG-D and RAG_T). (Please see 4th paragraph of the subsection 2.3 Procedures). Thank you.

#17 Procedures, lines 155-156: Has this approach been used to classify dominant and non-dominant previously? I am not convinced this is an ideal approach to use.

RESPONSE: We believe that the further clarification explained the approach more precisely. Text now reads: “For the purpose of this study, the performance on dominant and non-dominant sides were analyzed. To determine the dominant and non-dominant side, we first calculated the mean value for all B-cone performances (i.e. executions on the right side), and all A-cone performances (i.e. executions on the left side) for each participant and each executed test (e.g. CODS_D, CODS_T, RAG_D, and RAG_T). The performance side with lower mean value was determined as dominant side for each executed test (for each player separately). Please see last paragraph – page 4. Thank you.

#18 Statistical analysis, line 200: Do you mean "inter-testing" here rather than "intra-testing"?

RESPONSE: Yes, thank you. It is amended accordingly.

#19 Statistical analysis, line 208: What criteria was used to indicate a good CV? Please include.

RESPONSE: While this study included relatively complex tests, the CV < 10% was considered as appropriate. Text reads: “The ICC ≥ 0.75 and CV ≤ 10% were considered to indicate good (appropriate) reliability of studied CODS and RAG tests [13,34,35].” Thank you.

#20 Statistical analysis, lines 212-214: Please delete your hypothesis here as it is not needed.

RESPONSE: Deleted as suggested.

#21 Figure 2: The effect size for CODS TND should be a negative value (below zero) on this graph, given the team-level players performed better than top-level players. Please amend.

RESPONSE: Amended accordingly. Thank you. Please see Figure 2

#22 Discussion, line 291: I believe you mean "exception" rather than "execution" here?

RESPONSE: Corrected. Thank you!

#23 Discussion, line 309: Please explain the impact of a heterogenous sample of participants on reliability more directly here.

RESPONSE: The influence of “heterogeneity” of the sample on reliability is briefly explained. Text reads: Since heterogeneity of the sample increases the variability of the results, consequently increases the numerical values of the correlation coefficients (both between trials and/or measurements), and results in “better” reliability parameters [37].” (Please see 3rd paragraph of the Discussion. Thank you).

#24 Discussion, line 340: The reason of "test complexity" has been repeated numerous times up to here and seems quite repetitive. Can you include alternative suggestions or perhaps group your statements on test complexity together more to avoid duplication?

RESPONSE: We must agree that “test complexity” was accentuated in different places of the original manuscript. In this version we grouped different occurrences of similar explanations and tried to be more concise. Text now reads: “The futsal specific CODS tests displayed better reliability than RAG tests. Also, the protocols which involved dribbling were of lower reliability than those without dribbling. Finally, tests performed on dominant side were more reliable than those performed on non-dominant side. All of these differences in reliability are probably connected to the differences in text complexity (CODS vs. RAG, and with- vs. without-dribbling), and participants’ familiarity with performance and execution (dominant vs. non-dominant side). First, CODS tests are known to be less complex than RAG tests since during the RAG testing, there is a clear moment of surprise in which the athlete must promptly react to visual stimuli by immediately performing a proper movement pattern [13,38]. Second, tests which involve dribbling are more technically demanding than those performed without dribbling causes greater variations among trials and reduces the reliability [39]. Finally, the dominant-side in futsal is related to the side of the court the player is regularly positioned, and better motor-proficiency in specific directions, which altogether result in better stability of the performance and directly influences reliability (i.e. better reliability of tests performed on dominant side because of the better motor-proficiency) [40-42].“ (Please see 4th paragraph of the Discussion)

#25 Conclusions: It might be worth highlighting how many trials are needed during familiarisation before consistent results are seen in specific tests either in this section or in your Discussion more directly.

RESPONSE: In this version we specifically highlighted the problem of familiarization “from ouzr perspective”. Text reads: “The results of the study showed that newly developed CODS and RAG tests for futsal-specific agility testing were reliable. However, we may accentuate that proper reliability was achieved after specific familiarization of the players. Specifically, a day before testing all payers performed 2-3 familiarization trials at submaximal intensity in order to find the most appropriate movement template. Even after such familiarization, three maximal testing trials were necessary for the stabilization of the results.” (Please see 1st paragraph of the Conclusion; Thank you)

Staying at your disposal!

Authors

Reviewer 3 Report

You have conducted a well planned and well executed study! The Introduction is sufficient in information. In the Materials and Methods You could more in detail describe the randomization process, as the way You describe it, this is not a "true" randomization where everyone has a chance to be selected for every occasion. Also the selection of the 11 men subgroup for inter-testing reliability could have been more described. The Muscle Lab Timing gate I guess do have a product number. The degree of the "Y" shape should be noted, and also perhaps discussed, how much change in direction is wanted? The experts consulted have probably an opinion of this. The familiarization runs were of submaximal character, why not include one max effort trial also? Using Effect Size is a good thing, but You should state which one also in the text and not only in a figure legend. A reference to Cohen should also be included. In the Results, I would suggest that You use other words than "reached statistical..." (line 240 page 6) because this imply that (true enough) anything will get significant given a large enough N. The fact that the lowest-level performance group did out-perform the better group in one task could have been more emphasised in the Discussion. Also consider if two decimals are needed in Table 1 for mean and SD data, I think one decimal would be enough. Any way, be consistent, now there are inconsistencies. The Discussion is well presented.

Reference list: I have not detected any inconsistencies, but a thorough check is always warranted to see it the list adheres to the journal style. 

Some suggestions:

Line 40: add "successful" before "performance"

Line 64 ++: be consistent on how You report the source, either with publishing year in brakcets or not. (Bullock et al. (2012)... Pojskic et al. )

Author Response

REVIEWER 3

You have conducted a well planned and well executed study! The Introduction is sufficient in information.

RESPONSE: Thank you for recognizing the quality of our work. Also, thank you for your constructive comments and suggestions. Please see bellow how we amended the manuscript accordingly.

In the Materials and Methods You could more in detail describe the randomization process, as the way You describe it, this is not a "true" randomization where everyone has a chance to be selected for every occasion.

RESPONSE: Indeed, the study was not based on randomized approach. We tried to explain it more clearly in the revised version of the manuscript. Specifically, text reads: “Thirty-two male professional futsal players (age = 26.22±5.22 years; body height = 182.13±5.99 cm, body mass = 77.43±8.00 kg) from three futsal teams competing at the highest national level in Croatia voluntarily participated in the study.“ (please see first sentence of the subsection 2.2. Participants).

Also the selection of the 11 men subgroup for inter-testing reliability could have been more described.

RESPONSE: We agree that this issue deserved more attention. Text reads: “This study comprised repeated measurement design in order to define reliability of the newly developed tests. Additionally, throughout cross-sectional design we compared basic anthropometric variables, sprinting-, CODS- and RAG-performances from two groups of futsal players of varying performance-levels in order to identify construct validity of the applied tests. First, a randomly selected subgroup of 11 players performed the agility protocols on two separate testing occasions (e.g. test-retest) that were separated by 6 days to define inter-testing reliability. The intra-testing reliability was obtained from the results of all the participants involved in the study (n = 32). Finally, the groups based on performance (i.e., “team-level” and “top-level”; please see later for more details about dividing into performance-levels) were compared on dependent variables (body mass, body height, sprinting-, CODS- and RAG-performance).” (Please see 2nd paragraph of the Study design subsection). Thank you.

The Muscle Lab Timing gate I guess do have a product number.

RESPONSE: The product number is specified. “Core serial number: 08310013“. Thank you.

The degree of the "Y" shape should be noted, and also perhaps discussed, how much change in direction is wanted?

RESPONSE: In this version of the manuscript we specified all distances necessary for test reconstruction. Actually, “degrees” are hardly applicable if someone will try to reconstruct the test (e.g. 90 o; 38.66o and 51.44o degrees), while “triangle dimensions” are (Please see Figure 1a and 1b). Thank you.

The experts consulted have probably an opinion of this. The familiarization runs were of submaximal character, why not include one max effort trial also?

RESPONSE: Indeed, in the original version of the manuscript the consultations with the futsal experts was only briefly explained. In the revised version we tried to be more accurate. Text reads: “In the initial phase of the investigation the authors of the study collaborated with futsal experts to gather opinions regarding futsal-specific agility. The authors presented the basic idea, technical details, and theoretical boundaries of test execution (i.e. available equipment) and asked experts to express their opinion about the most appropriate scenario which will be convenient for the purpose of testing of the futsal specific CODS and RAG for all field players (e.g. defenders, wingers, and pivots). The applied knowledge obtained allowed development of relatively simple testing procedures that evaluated different agility components of the sport (please see “Procedures” section below for details)” (Please see first paragraph of the 2.1. Study design. Thank you!)

Using Effect Size is a good thing, but You should state which one also in the text and not only in a figure legend. A reference to Cohen should also be included.

RESPONSE: Thank you for noticing it. The origin of the ES is mentioned in the text also, and reference is included. Text reads: 2 „Additionally, effect sizes (ES; Cohen’s d) for differences in sprinting- and agility-performances between performance-group were calculated (difference in group-means divided by the pooled standard deviations), and interpreted using the following qualitative descriptors: < 0.2 as trivial, 0.21-0.49 as small, 0.50-0.79 as medium, >0.79 as strong [35].”

In the Results, I would suggest that You use other words than "reached statistical..." (line 240 page 6) because this imply that (true enough) anything will get significant given a large enough N.

RESPONSE: Amended accordingly. Text reads: “Significant ANOVA effects were evidenced for RAG_DND, and post-hoc analysis revealed significant differences between all three trials for this test procedure.” (Please see 2nd paragraph of the Results).

The fact that the lowest-level performance group did out-perform the better group in one task could have been more emphasised in the Discussion.

RESPONSE: In this version we briefly discussed the finding of better CODS_TND in team players. tet reads: “The fact that team-level-players outperformed top-level-players in CODS_TND is not surprising since this performance is probably not so related to skill-levels. Also, performance-groups did not differ in sprint capacity, while it is possible that team-level players are superior in other conditioning capacities known to be important determinants of CODS (i.e. jumping-, power-capacity) [47], which should be investigated in more details in future studies. “ (please see 7th paragraph of the Discussion) Thank you.

Also consider if two decimals are needed in Table 1 for mean and SD data, I think one decimal would be enough. Any way, be consistent, now there are inconsistencies.

RESPONSE: The means and standard deviations in Table 1 are now presented with one decimal place. Thank you.

The Discussion is well presented.

RESPONSE: Thank you!

Reference list: I have not detected any inconsistencies, but a thorough check is always warranted to see it the list adheres to the journal style.

RESPONSE: In this version of the manuscript References are thoroughly checked for consistency according to Journal’s style. Thank you.

Some suggestions:

Line 40: add "successful" before "performance"

RESPONSE: Added as suggested.

Line 64 ++: be consistent on how You report the source, either with publishing year in brakcets or not. (Bullock et al. (2012)... Pojskic et al. )

RESPONSE: Corrected, thank you.

Staying at your disposal

Authors

Round 2

Reviewer 2 Report

Thank you for addressing the extensive revisions provided in the previous submission. I believe all major concerns have been fixed and a stronger manuscript has been presented in this submission. I have no further major revisions to request.